# A prospective randomized trial on abacavir/lamivudine plus darunavir/ritonavir or raltegravir in HIV-positive drug-naïve patients with CD4<200 cells/uL (the PRADAR study)

Cristina Mussini[1]*, Enrica Roncaglia[1], Vanni Borghi[1], Stefano Rusconi[2], Silvia Nozza[3], Anna Maria Cattelan[4], Daniela Segala[5], Paolo Bonfanti[6], Antonio Di Biagio[7], Enrico Barchi[8], Emanuele Focà[9], Anna Degli Antoni[10], Stefano Bonora[11], Daniela Francisci[12], Silvia Limonta[2], Andrea Antinori[13], Gabriella D'Ettorre[14], Franco Maggiolo[15]

1 Clinic of Infectious Diseases, University of Modena and Reggio Emilia, Modena, Italy, 2 Clinic of Infectious Diseases, DIBIC Luigi Sacco, University of Milan, Milan, Italy, 3 Clinic of Infectious Diseases, University Vita e Salute, San Raffaele Hospital, Milan, Italy, 4 Clinic of Infectious Diseases, Padova Hospital, Padova, Italy, 5 Clinic of Infectious Diseases, Sant'Anna Hospital, Ferrara, Italy, 6 Department of Infectious Diseases, Lecco Hospital, Lecco, Italy, 7 Clinic of Infectious Diseases, San Martino Hospital, Genova, Italy, 8 Department of Infectious Diseases, Santa Maria Nuova Hospital, Reggio Emilia, Italy, 9 Clinic of Infectious Diseases, University of Brescia and ASST Spedali Civili, Brescia, Italy, 10 Clinic of Infectious Diseases, University of Parma, Parma, Italy, 11 Clinic of Infectious Diseases, University of Torino, Torino, Italy, 12 Clinic of Infectious Diseases, University of Perugia, Perugia, Italy, 13 National Institute for Infectious Diseases L. Spallanzani, Rome, Italy, 14 Clinic of Infectious Diseases, University La Sapienza, Rome, Italy, 15 Department of Infectious Diseases, Bergamo Hospital, Bergamo, Italy

* crimuss@unimore.it

## Abstract

### Background

Very few data are available on treatment in HIV Late presenter population that still represents a clinical challenge.

### Methods

Prospective, multicenter, randomized open-label, 2 arm, phase-3 trial comparing the 48-week virological response of two different regimens: abacavir/lamivudine + darunavir/r vs abacavir/lamivudine + raltegravir in antiretroviral naive with CD4+ counts < 200/mm3 and a viral load (VL)<500,000 copies/mL. The primary Endpoint was the proportion of patients with undetectable viremia (VL<50 copies/mL) after 48 weeks. The planned sample size for this trial was 350 patients.

### Results

In 3 years, 53 patients were screened and 46 enrolled: 22 randomized to raltegravir and 24 to darunavir/r; 7 patients were excluded, 4 because of a VL >500,000 copies/mL and 3 for HLAB5701 positivity. The snapshot analysis at 48 weeks showed a virologic success of 77.3% in raltegravir and 66.7% in darunavir/r. Time to starting treatment was 34.5 days in

**Data Availability Statement:** All relevant data are within the manuscript and its Supporting Information files.

**Funding:** Funding for this study was provided by ViiV Healthcare, the authors are solely responsible for final content and interpretation. Supported in part by a research grant from Investigator-Initiated Studies Program of Merck Sharp & Dohme Corp. The opinions expressed in this paper are those of the authors and do not necessarily represent those of Merck Sharp & Dohme Corp. Supported in part by a research grant by Janssen. The opinions expressed in this paper are those of the authors and do not necessarily represent those of Janssen.

**Competing interests:** We received unrestricted financial support for the study from the Companies: ViiV Healthcare, Merck Sharp and Dohme Corp and Janssen, but this does not alter our adherence to PLOS ONE policies on sharing data and materials.

raltegravir and 53 days in darunavir/r. At the as treated analysis, the median CD4 counts at 48 weeks was 297 cells/μL in raltegravir and 239 cells/μL in darunavir/r. No difference in total cholesterol, while triglycerides were higher in the darunavir/r arm. No statistical analyses were performed due to the low number of patients enrolled.

## Conclusions

Late presenter patients are frequent but very difficult to enroll in clinical trials, especially in western countries. These regimens and the conditions of many patients could not allow the test and treat strategy. The rate of virologic success was higher than 65% in both arms with a median CD4 cell count >200/μL at week 48.

## Trial registration

EUDRACT number: 2011-005973-21

## Introduction

Nowadays, late presentation represents one of the major obstacles to HIV eradication. Indeed, after more than 30 years since the beginning of the epidemic, around 40% of new diagnoses worldwide still occur in subjects with less than 200 CD4+/uL [1,2]. Reasons for late presentation range from perception of not being at risk for HIV infection, access to services, stigma and discrimination and are difficult to be addressed [3]. Concerning antiretroviral treatment, patients presenting with an advanced HIV disease, especially if they are symptomatic or with a diagnosis of one or more opportunistic infections, are usually excluded from randomized clinical trials and only a few studies have investigated antiretroviral regimens in these patients [4–6]. Actually, since most studies enrolled a small number of these patients, guidelines do not give any specific treatment indications on how to treat this population. The first question was when to start treatment in these patients especially in presence of an opportunistic infection and some randomized clinical trial have shown that, with the exception of cryptococcal and tuberculosis meningitis, it is better, from a prognostic point of view, to start treatment earlier [7–11]. The second question, since clinicians perceive advanced patients as more fragile, was with how many drugs treatment should be started. Recently, 2 trials have been conducted in the attempt to understand if 4 drugs could achieve better results than 3. Indeed, no intensification approach either with maraviroc or with raltegravir reached a higher rate of suppression than triple therapy [12,13]. The third question concerns which third drug should be used in triple combinations. An underpowered study showed that among protease inhibitors, darunavir/r reached better virological results than atazanavir/r in advanced naïve patients [14]. Concerning integrase inhibitors, the most widely studied in this population, even if not in specific trials, is raltegravir, since either StartmRk or ACTG 5257 enrolled a not negligible percentage of advanced subjects [15,16]. Concerning the backbone, no specific data are available. Aim of the present study, which was designed as a large open label trial, was to evaluate the virological efficacy of two drug regimens: abacavir/lamivudine plus either raltegravir (RAL) 400 mg twice a day or darunavir/r (DRV/r) 800/100 mg once a day in subjects with a diagnosis of HIV infection, CD4 count <200 cells/uL, and HIV-RNA <500,000 cp/mL.

## Methods

A prospective, multicenter, randomized open-label, 2 arm, phase-3 trial comparing the 48-week virological response of two different regimens: abacavir/lamivudine + DRV/r vs abacavir/lamivudine + RAL in antiretroviral naive HIV+ individuals, HLA B5701 negative, presenting for care with CD4+ cell count < 200/mm3 and a viral load (VL)<500,000 copies/mL. Primary Endpoint: Proportion of patients with undetectable viremia (VL <50 copies/mL) after 48 weeks, secondary Endpoints: Change in CD4+ cell count from baseline through week 48 and time to virological rebound. Abacavir/lamivudine was chosen even if resulted inferior to tenofovir disoproxil fumarate /emtricitabine in the ACTG5202, since no data were available in combination either with raltegravir or darunavir [17]. Considering the results of the ACTG 5202 we decided to include only patients with a baseline viral load <500,000 copies/mL.

### Inclusion criteria

Males or females antiretroviral-naive aged 19–65 years HIV-1 antibody seropositive, with a CD4+ cell count <200 cells/uL, an HIV RNA level <500,000 copies/mL and able to provide written informed consent. All patients should be HLA B57 or HLA B5701 negative and they should not have resistance to any study drug at randomization. Patients with an opportunistic infection were included as long as this was diagnosed more than 2 weeks prior to screening. Patients must had met the following laboratory criteria: neutrophil count > 1,000 cells/mm$^3$, haemoglobin > 9.0 grams/dl (men and women), platelet count $\geq$ 75,000 cells/mm$^3$, alkaline phosphatase < 3.0 the upper limit of normal, ALT and AST < 3.9 times the upper limit of normal, total bilirubin > 1.5 times the upper limit of normal. Female patients of childbearing potential must be willing to use a reliable form of contraception, which will include a medically approved form of barrier contraception.

### Exclusion Criteria

Patients with the following characteristics were excluded: a positive HLA B57 or B5701, chronic B hepatitis, the presence of genotypic mutations for any of the study drugs, an opportunistic infection diagnosed less than 2 weeks before screening, an HIV RNA level >500,000 copies/mL, pregnant or breastfeeding, with a current drug, alcohol or substance abuse. Finally, patients receiving any investigational drug or anti-neoplastic radiotherapy/chemotherapy other than local skin radiotherapy within 12 weeks of starting medication were excluded.

### Planned sample size

As this was a non-inferiority trial, we wanted to calculate the difference in the proportions of patients experiencing the primary outcome in the two treatment arms and calculated a 95% confidence interval for this. Non-inferiority of the RAL arm should have been demonstrated if the lower limit of the 95% confidence interval was greater than -12%. In case non-inferiority would have been met, analyses for superiority would have been performed.

The planned sample size for this trial was 350 patients. Assuming an underlying response rate of 80% in each of the arms at week 48, this sample size (175 patients per arm) would have provided 80% power to demonstrate non-inferiority of the RAL arm compared to the DRV/r arm (alpha = 0.025) with a non-inferiority margin of 12%. Unfortunately, we could not reach the number of patients planned.

## Statistical analyses

Descriptive results are presented as means ± standard deviation (SD), medians with interquartile range (IQR), and percentages with 95% confidence intervals (CI). Inferential statistics using either parametric or non-parametric tests are used, as appropriate. Chi-square or Fisher's exact test are used to analyze categorical variables. Paired T-test and Wilcoxon signed rank test are used to analyze within participant differences at different time points. Analysis was performed with SPPS for windows 17.0.

## Ethical issues

The study was approved by the ethical committee of the coordinating centers and those of all participant centers.

## Results

Between February 2012 and February 2016, 53 patients were screened and 46, after having signed the informed consent, were randomly allocated in the two groups (22 with RAL and 24 with DRV/r). 7 patients were excluded, 4 because of a VL >500,000 copies/mL and 3 for HLAB5701 positivity (Fig 1).

Baseline characteristics were fairly balanced between groups (Table 1), although patients in the RAL group were younger and with a slightly more advanced CDC stage. Opportunistic infections and pathologies were oral-esophageal candidiasis (5 in RAL and 6 in DRV/r); *Pneumocystis jirovecii* pneumonia (1 in RAL and 3 in DRV/r) and neuro-toxoplasmosis (one in each group). Kaposi Sarcoma, cryptococcal meningitis, lymphoma and wasting syndrome (one each) were diagnosed only in the RAL group, while Cytomegalovirus disseminated disease (2 cases) only in the DRV/r group. Two patients in the DRV/r group were diagnosed with two opportunist infections (*Pneumocystis jirovecii* pneumonia and Cytomegalovirus disseminated disease). Although the median time between HIV diagnosis and cART start was similar between groups (34.5 days for RAL and 53 days for DRV/r), in the latter case, we observed a much greater interquartile range due to a few patients whose therapy was delayed.

Beside antiretroviral therapy, due to their advanced status, patients received other concomitant therapies. In the DRV/r group concomitant therapies ranged from 0 to 14, while in the RAL group the range was from 0 to 15. Most commonly used drugs (> 3 patients) were antibiotics (15 cases in each group); antifungals (7 in DRV/r and 5 in RAL); antivirals (6 in DRV/r and 4 in RAL); non -steroids anti-inflammatory drugs (3 in DRV/r and 10 in RAL); steroids (3 in DRV/r and 5 in RAL); vitamins (7 in DRV/r and 8 in RAL); gastro-enteric drugs (5 in DRV/r and 9 in RAL and CNS drugs (5 in DRV/r and 6 in RAL).

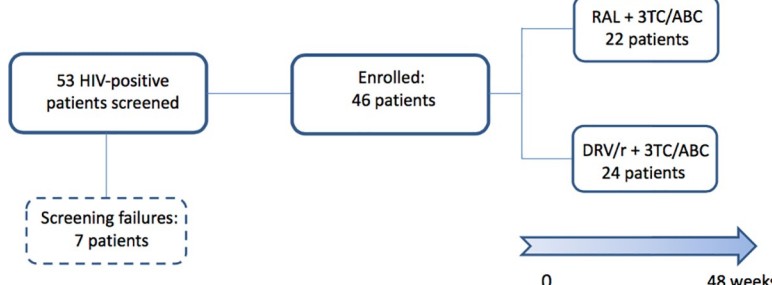

**Fig 1. Patients' disposition of the study.** 53 patients were screened in the present study: 46 patients were randomly allocated in the two groups (22 with RAL and 24 with DRV/r) and 7 patients were excluded (4 because of a VL >500000 copies/mL and 3 for HLAB5701 positivity).

**Table 1. Baseline characteristics of patients according to the third agent.**

|  | raltegravir | Darunavir/ritonavir |
|---|---|---|
| Gender, male/female, number (%) | 19/3 (86.4/13.6) | 19/5 (79.2/20.8) |
| Age, years, median (IQR) | 41 (32.5–45.5) | 35 (30–46) |
| Risk factor for HIV, number (%) |  |  |
| MSM | 9 (41.0) | 11 (45.9) |
| Heterosexual contacts | 11 (50.0) | 10 (41.8) |
| IVDU | 1 (4.5) | 1 (4.1) |
| Other | 1 (4.5) | 1 (4.1) |
| Unknown | 0 | 1 (4.1) |
| CDC stage, number (%) |  |  |
| A | 10 (45.5) | 15 (62.5) |
| B | 5 (22.7) | 3 (12.5) |
| C | 7 (31.8) | 6 (25.0) |
| HIV-RNA, copies/ml, Median (IQR) | 89731 (54319–153675) | 112250 (71316–275554) |
| CD4, cells/mcL, median (IQR) | 108 (44–172) | 107 (35–170) |
| CD8, cells/mcL, median (IQR) | 629 (352–992) | 771 (562–1068) |

Note: MSM, men who have sex with men; IVDU, intravenous drug users; CDC, Center for Diseases Control

According to a snapshot algorithm, on the ITT population, after 48 weeks the proportion of treatment success was 77.3% in the RAL group and 66.7% in the DRV/r group. The 95% Confidence interval for the difference between these proportions was from -15.4% to 36.4%. No difference was found considering the rate of suppression below 200 copies/mL. Virologic failure was observed in 22.7% of patients treated with RAL and 29.2% of those receiving DRV/r while only one patient on DRV/r (4.2%) did not have data in window. According to the as treated analysis virologic response was faster in the RAL group being the proportion of subjects below a 50 copies/ml threshold 0f 22% at 4 weeks, 70% at 24 weeks and 94.1% at 48 weeks; the same figures in the DRV/r group were 4.8%; 40% and 84.2%

At the end of follow up, the median CD4+ cell count raised to 297 cell/uL (IQR 218–454) in the RAL arm and to 239 cell/uL (IQR 182–458) in the DRV/r group. The increment was slightly higher in the RAL arm (189 vs 112 cells/uL). On the contrary, the median CD8+ cell count raised less in the RAL arm (w48 value 809 cells/uL; IQR 662–1476) than in the DRV/r arm (w48 value 1172 cells/uL; IQR 857–1485). Consequently, these differences influenced the dynamics of the CD4+/CD8+ ratio that increased slightly more in the RAL arm. At baseline the median CD4+/CD8+ ratio was 0.12 (IQR 0.08–0.17) for DRV/r and 0.15 (IQR 0.06–0.22) for RAL while the same values were respectively 0.30 (IQR 0.18–0.37) and 0.34 (IQR 0.19–0.64) at 48 weeks.

We did not find clinically significant changes, from baseline to week 48, in hematological exams, hepatic markers, LDH and CPK, nor grade 3–4 laboratory abnormalities for any considered parameter. Renal function remained stable over time with creatinine levels increasing from 0.8 (IQR 0.0.74–0.89) to 0.9 (IQR 0.83–0.96) in the DRV/r arm and from 0.8 (IQR 0.0.68–0.96) to 0.9 (IQR 0.75–0.92) in the RAL arm. Changes in lipids were also small, with limited median increments in total cholesterol, LDL cholesterol and triglycerides. However, especially for LDL cholesterol and triglycerides the increment was slightly more evident in the DRV/r arm (Fig 2). In the DRV/r arm a desired level of LDL cholesterol < 130 mg/dl was observed only in 5/15 patients at week 48, while the same proportion in the RAL arm was 10/16 (P = 0.023). Similarly, a desired triglycerides level < 170 mg/dl was observed in 10/16 DRV/r patients and in 12/17 RAL patients (P = 0.902).

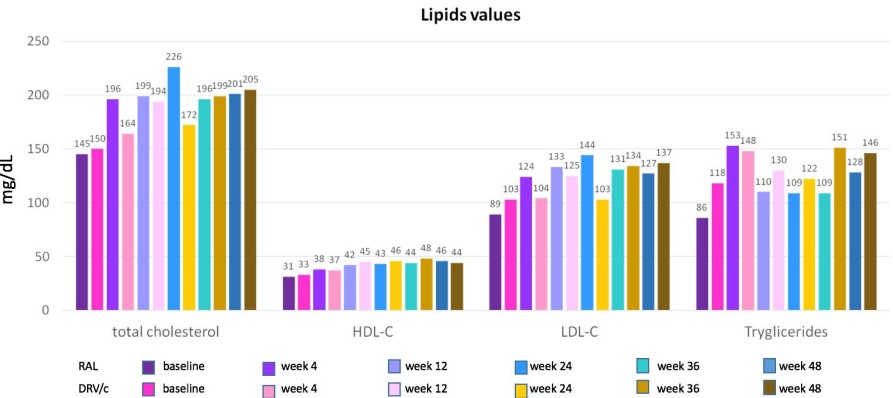

**Fig 2. Lipid values in the RAL and DRV/r patients.** Lipid values, expressed as mg/dl. A desired level of LDL cholesterol < 130 mg/dl was observed only in 5/15 patients at week 48 in the DRV/r arm, while the same proportion in the RAL arm was 10/16 (P = 0.023). A desired triglycerides level < 170 mg/dl was observed in 10/16 DRV/r patients and in 12/17 RAL patients (P = NS).

Two patients (one in each group) stopped treatment because of adverse events: acute renal failure in the RAL arm and allergic reaction in the DRVr/r group.

Beside these two cases, several other patients reported adverse events not leading to treatment discontinuation during the treatment period. In the DRV/r arm 19/24 patients reported between 1 and 7 adverse events, in the RAL arm 11/22 patients reported between 1 and 4 adverse events (Table 2). Finally, a pregnancy was observed in the RAL arm, but the woman continued the study and gave birth to a healthy child.

## Discussion

The first evidence from the present study is that it is very complicated to conduct clinical trial in advanced HIV-infected patients especially in western countries where many therapeutic options are available. Indeed, due to the low number of subjects enrolled we did not feel confident in performing any comparative analysis between RAL and DRV/r in regards to virologic or immunologic response. Actually, even if the use of abacavir/lamivudine could represent a limit for a rapid enrollment due to the test for HLAB5701 and the threshold of viral load <500,000 copies/mL, we did not expect to have such a low number of patients included in 3 years. Actually, Italy, due to the early day HIV epidemiology *i.e.* former drug users and heterosexual contacts, is among the countries with a higher rate of late presentation—around 56% of new diagnosis–[18] and the inclusion criteria, as shown by our results, played only a minor role in limiting the number of enrollments. Thus, we decide not to modify our inclusion criteria and/or increase the number of participating clinical centers. Possible reasons for this low

**Table 2. Adverse events reported in more than 2 patients (number and percent).**

| Adverse event | raltegravir | Darunavir/ritonavir |
|---|---|---|
| Rash/dermatitis | 4 (18%) | 6 (25%) |
| Diarrhea | 1 (4.5%) | 4 (16%) |
| Pneumonia/bronchitis | 5 (22%) | 3 (12%) |
| Oral candidiasis | 2 (9%) | 2 (8%) |
| Sore throat | 1 (4.5%) | 3 (12%) |
| Herpes | 2 (9%) | 3 (12%) |
| Flu-like syndrome | 2 (9%) | 1 (4%) |

rate of enrollment could be: firstly, the study started when, after the results of ACTG 5202, abacavir was perceived by clinicians as less potent from a virological point of view than tenofovir disoproxil fumarate [17], secondly, clinicians had some doubts in randomizing advanced patients. Actually, this was not a problem only of our study, since this happens also in other countries with high rate of late presentation as Spain, where only small studies in this population were conducted, also changing the end-points from virological success to immunological changes [5,6]. More importantly, our results show that even when patients are enrolled they are difficult to be treated. First, the level of suppression reached in both arms is very far from that we are recently used to obtain in non-advanced subjects *i.e.* almost 90% [19]. DRV/r and RAL were chosen since they were both recommended first line treatment and there were no data in combination with abacavir/lamivudine at the time the trial was designed. Concerning RAL the rate of suppression at 48 weeks in the STARTMRK trial in naïve subjects, 47% with a CD4 count<200 cells/uL, in combination with a TDF/FTC backbone was 86% [15]. During the period of enrollment more data became available on RAL use in advanced patients and the results could have had a negative impact on clinicians regarding enrolling patients in the study. Indeed, the SHIELD study, a pilot study on 35 naïve subjects receiving abacavir/lamivudine + RAL showed a level of suppression of 91% [20], while a subgroup analysis of the SPRING-2 trial showed a rate of virologic success in subjects with <200 CD4 cells/uL of 68% for RAL and 78% for dolutegravir [21]. RAL-containing HAART was also tested in the NEAT001 trial as an initial dual regimen combined with DRV/r [22]. The RAL-containing arm did not perform as well as the standard-of-care arm containing darunavir/ritonavir plus two N(n)RTI in the subgroup of patients with a CD4 cell count <200/μL. As well in the Gemini 1 and 2 trials, a lower response in the DTG plus 3TC group than in the three-drug regimen group was observed in the subgroup of participants with baseline CD4+ count ≤200 cells/μL [23]. Concerning DRV/r in naïve subjects the rate of suppression at 48 weeks in the FLA-MINGO trial was 83% [24]. These levels of suppression are related to the whole population since when we examine the subgroups with viral load>100,000 copies/mL or CD4<200 cells/ uL the percentages of subjects who reached an undetectable viral load are lower; for example, DRV/r among late presenters in the IMEA 040 DATA trial showed an 80% rate of suppression [14].

Even if our data could not be compared to large randomized trial, it is evident that the virological results are mainly due to the characteristics of the patients rather than to the drug chosen. It is possible that more recent combination will lead to better results. Actually, in a subgroup analysis of the GS1490, dolutegravir and bictegravir obtained, in combination with tenofovir alafenamide/emtricitabine (TAF/FTC) a higher rate of success: 100 and 95%, respectively [19]. Second, the time before treatment was started. The START trial showed how relevant for the single patients is to start antiretroviral therapy as soon as possible in order to avoid AIDS events [25]. In the most recent time, the period between diagnosis and treatment has become even shorter since in the attempt to increase adherence some groups have started to evaluate a test and treat approach [26,27]. Patients enrolled in our study started treatment after more than one month. The reasons for this delay were related first to the clinical conditions of the subjects since our patients presented a variety of opportunistic infections and in some cases, more than one. Indeed, in patients presenting with opportunistic infections as in this study, many trials have been conducted on the correct timing to start antiretroviral therapy balancing efficacy and IRIS. ACTG 5164 showed that in *Pneumocystis jirovecii* pneumonia is better to start after a median of 12 days, while 2 randomized trials conducted in cryptococcal meningitis showed that starting earlier could increase mortality [7,10,11]. Concerning tuberculosis (TB) starting early could be relevant for mortality only in presence of a CD4+ cell count <50 cells/uL, while in TB meningitis starting earlier led to a greater incidence of side

effect without no survival benefit [8,9]. Second, the regimens studied. Indeed, the use of abacavir due to the HLA B5701 test represents *per se* a delay and the association of raltegravir and abacavir/lamivudine, with a low genetic barrier to resistance requires the results of baseline genotyping tests. Actually, the LAPTOP study comparing in late presenters bictegravir/tenofovir alafenamide/emtricitabine and darunavir/cobicistat/ tenofovir alafenamide/emtricitabine, both regimens with a high genetic barrier and without abacavir will allow us to understand which is the time to starting treatment in this specific population [28]. The information will be very relevant in clinical practice since subjects who will be included in the LAPTOP will be even more difficult to treat than that of this study since we have included symptomatic and asymptomatic with a CD4 count <200 cells/uL, while to be enrolled in the LAPTOP subjects should have AIDS with any CD4 cell count or severe bacterial infection, but with a CD4 cell count < 200/μl within 30 days prior to study entry or be asymptomatic with CD4 cell count < 100/μL within 30 days prior to study entry and an entry HIV viral load > 1000 copies/mL or currently receiving treatment for opportunistic infection. Another finding that indicates the difficult management of these patients is the high number of concomitant medications. Indeed, while in non-advanced naïve patients we still discuss on the role of a single tablet regimen compared to more pills, the picture in late presenters is completely different [29]. Moreover, this high number of co-medications and the complex clinical picture could explain the onset of side effects that are really difficult to ascribe to a single antiretroviral drug.

Third, there was no real surprise concerning lipid profile and viro-immunological effects. As expected, on the basis of the results of ACTG 5257, RAL showed a better profile than DRV/r and also a better immunological profile since subjects receiving RAL showed a higher CD4 cell increase and a higher decrease of CD8 cells. This last datum confirms the effect of RAL on CD4/CD8 ratio as shown in the STARTMRK [30].

In conclusion, despite the low number of patients enrolled our study has confirmed how difficult is to treat patients with advanced HIV disease. Actually, all the rules that we are using in non-immunocompromised naïve patients as early treatment or simple regimens are difficult to apply in patients with late presentation, especially if diagnosed with opportunistic infections. Actually, more data are needed and we hope that a large prospective trial as the LAPTOP will answer some unsolved questions.

## Supporting information

**S1 File. CONSORT 2010 checklist.**
(DOC)

**S2 File. RTLP_vers 1.2 def.** Original Protocol.
(DOC)

## Author Contributions

**Conceptualization:** Cristina Mussini, Stefano Rusconi, Silvia Nozza, Antonio Di Biagio, Andrea Antinori, Franco Maggiolo.

**Data curation:** Enrica Roncaglia, Vanni Borghi, Silvia Limonta.

**Formal analysis:** Franco Maggiolo.

**Funding acquisition:** Cristina Mussini.

**Investigation:** Cristina Mussini, Stefano Rusconi, Silvia Nozza, Anna Maria Cattelan, Daniela Segala, Paolo Bonfanti, Antonio Di Biagio, Enrico Barchi, Emanuele Focà, Anna Degli

Antoni, Stefano Bonora, Daniela Francisci, Silvia Limonta, Andrea Antinori, Gabriella D'Ettorre, Franco Maggiolo.

**Methodology:** Cristina Mussini.

**Project administration:** Cristina Mussini.

**Software:** Vanni Borghi, Franco Maggiolo.

**Writing – original draft:** Cristina Mussini, Stefano Rusconi, Franco Maggiolo.

**Writing – review & editing:** Enrica Roncaglia, Vanni Borghi, Silvia Nozza, Anna Maria Cattelan, Daniela Segala, Paolo Bonfanti, Antonio Di Biagio, Enrico Barchi, Emanuele Focà, Anna Degli Antoni, Stefano Bonora, Daniela Francisci, Silvia Limonta, Andrea Antinori, Gabriella D'Ettorre.

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
