## [Decision Letter · Decision Letter 0]

19 Jul 2019

PONE-D-19-14981

A Prospective Randomized trial on Abacavir/lamivudine plus DArunavir/ritonavir or Raltegravir in HIV-positive drug-naïve patients with CD4<200 cells/uL (the PRADAR Study)

PLOS ONE

Dear Dr. Mussini,

Thank you for submitting your manuscript to PLOS ONE. After careful consideration, we feel that it has merit but does not fully meet PLOS ONE’s publication criteria as it currently stands. Therefore, we invite you to submit a revised version of the manuscript that addresses the points raised during the review process.

In addition to addressing the comments from the reviewers, please could you review the following:

1.  In the abstract it is stated 'the test and treat strategy is rarely applicable'; is this related to the study design and the use of ABC/3TC or RTG and do you think this strategy will become more applicable with modern ART and the strategy of the LAPTOP study? This should be revised.

2.  In the discussion section, please consider mentioning the LAPTOP study and how the approach differs from your study.

We would appreciate receiving your revised manuscript by Sep 02 2019 11:59PM. To enhance the reproducibility of your results, we recommend that if applicable you deposit your laboratory protocols in protocols.io, where a protocol can be assigned its own identifier (DOI) such that it can be cited independently in the future. For instructions see: http://journals.plos.org/plosone/s/submission-guidelines#loc-laboratory-protocols

We look forward to receiving your revised manuscript.

Kind regards,

Alan Winston

Academic Editor

PLOS ONE

Journal Requirements:

2) We note that your study protocol contains a confidentiality notice. As detailed in PLOS Editorial Policy, study protocols must be published alongside papers reporting clinical trials in the event of acceptance. Please note that, should your paper be accepted, all content including the protocol will be published under the Creative Commons Attribution (CC BY) 4.0 license, which means that it will be freely available online, and any third party is permitted to access, download, copy, distribute, and use these materials in any way, even commercially, with proper attribution. In order to publish any previously copyrighted material, PLOS ONE requires permission from the original copyright holder of the content to publish it under the CC BY 4.0 license.

Before we proceed with your submission, please provide written permission to publish the protocol under the Creative Commons Attribution (CC BY) 4.0 license. You may provide a written statement from the copyright owners (if they are not the authors of the manuscript) or forward your email correspondence indicating that you have been granted this permission. Additionally, please send a clean copy of the protocol with the confidentiality notice removed.

3)  Thank you for submitting your clinical trial to PLOS ONE and for providing the name of the registry and the registration number. The information in the registry entry suggests that your trial was registered after patient recruitment began. PLOS ONE strongly encourages authors to register all trials before recruiting the first participant in a study.

a) your reasons for your delay in registering this study (after enrolment of participants started);

b) confirmation that all related trials are registered by stating: “The authors confirm that all ongoing and related trials for this drug/intervention are registered”.

Please also ensure you report the date at which the ethics committee approved the study as well as the complete date range for patient recruitment and follow-up in the Methods section of your manuscript.

4)  Thank you for stating the following financial disclosure:

 [Funding for this study was provided by ViiV Healthcare, the authors are solely

responsible for final content and interpretation.

Supported in part by a research grant from Investigator-Initiated Studies Program of

Merck Sharp & Dohme Corp. The opinions expressed in this paper are those of the

authors and do not necessarily represent those of Merck Sharp & Dohme Corp.

Supported in part by a research grant by Janssen. The opinions expressed in this

paper are those of the authors and do not necessarily represent those of Janssen.].               

5) Thank you for stating the following in the Competing Interests section:

[The authors have declared that no competing interests exists].

We note that you received funding from a commercial source: ViiV Healthcare, Merck Sharp & Dohme Corp and Janssen.

Please provide an amended Competing Interests Statement that explicitly states these commercial funders, along with any other relevant declarations relating to employment, consultancy, patents, products in development, marketed products, etc.

6) Please include captions for your Supporting Information files at the end of your manuscript, and update any in-text citations to match accordingly. Please see our Supporting Information guidelines for more information: http://journals.plos.org/plosone/s/supporting-information.

Reviewers' comments:

Reviewer's Responses to Questions

**Comments to the Author**

1. Is the manuscript technically sound, and do the data support the conclusions?

Reviewer #1: Yes

Reviewer #2: Yes

Reviewer #3: No

Reviewer #4: Partly

2. Has the statistical analysis been performed appropriately and rigorously? 

Reviewer #1: Yes

Reviewer #2: Yes

Reviewer #3: Yes

Reviewer #4: No

3. Have the authors made all data underlying the findings in their manuscript fully available?

Reviewer #1: Yes

Reviewer #2: Yes

Reviewer #3: Yes

Reviewer #4: Yes

4. Is the manuscript presented in an intelligible fashion and written in standard English?

Reviewer #1: Yes

Reviewer #2: Yes

Reviewer #3: No

Reviewer #4: Yes

5. Review Comments to the Author

Reviewer #1: There are only a few randomized clinical trials on antiretroviral head-to-head comparison between protease-inhibitor (PI) versus integrase-inhibitor (INSTI) based combination triple therapy. PRADAR was a multicentre, national (Italian), open label study and an attempt to fill this important data gap. The study design radically addressed the open question, whether standard combination ART should better contain an INSTI, or “best practice-PI”. Because the chosen study population was moreover the difficult-to-treat HIV-late presenters, with very low first measured CD4-cell counts (<200/µL), the study finally failed by trying to reach “a bridge too far”. Though the attempt was heroic, this report’s conclusions should not be in vain, as this may guide future RCTs to answer the yet open question.

The main problem of this study was the patient recruitment underperformance, reflected by the fact that only 46 individuals were randomized in three years – far from trial’s goal of 350. There are different reasons for this failure: apart from the challenging study population, that may often be deterred from a trial participation due to many reasons, there was another game changer occurring in the middle of the recruitment time – the publication of the START-study, as discussed. All antiretroviral therapy guidelines were subsequently changed to prompt treatment initiation; this diminished indirectly the potential study population.

The study addresses a major important study question - better INSTI or PI for late presenters? It anticipated the study question long before other major trials, i.e. currently under investigation (see NCT03696160). Though the report did not fully succeed, it should not hide its’ light under the bushel. Therefore, I consider the report on PRADAR as valuable and worth reading. Nevertheless, I recommend to consider the following minor issues:

1. Discussion, page 18/30, line 3, following, you wrote: “Indeed, due to the low number of subjects enrolled it was not possible to perform any comparative analysis between RAL and DRV/r.” – But you have performed comparisons! And you had interesting findings, e.g. on clinical adverse events and cholesterol differences between groups. – Please clarify.

2. Discussion, page 19/30, first lines/first para – please consider to discuss the following facts: in NEAT001-randomized clinical trial (NCT01066962), the raltegravir-containing arm did not perform as well as the standard-of-care arm containing darunavir/ritonavir plus two nucleosidal reverse transcriptase inhibitors, in the subgroup of patients with a CD4 cell count <200/µL. Similar findings were detected in patients with another actual dual regimen in naives: NCT02831764 & NCT02831673.

3. Discussion, page 19/30, lines 14 & following – see above remarks on START study findings release while PRADAR study recruitment.

4. Discussion, page 20/30, second para – you showed differences in week 48-outcomes of complete HIV-RNA suppression (<50 copies/mL) between arms, i.e. the primary end point (94.3 vs. 84.2%). Please consider to argue on the differences concerning the 50 copies/mL-threshold for PI vs. INSTI drug classes and to display as well the results for a 200 copies/mL-threshold, as the ACTG recommended this different limit of detection for studies on protease inhibitors.

5. Figure 1, page 27/30: please consider to include the reasons for 7 screening failures.

Reviewer #2: Good work by the authors. I agree it is difficult study to recruit to in the current environment. many other options are available and perceived as superior to the combinations you are trying to test.

in Your abstract ( in the last line) you stated: the rate of virologic success is similar to that described in the literature and very far from results of the recent trials in naïve patients.

this statement needs to be revisited in my view and quantified.

in the last line of your introduction: >500000 should be <500000.

Reviewer #3: • The paper by Mussini et al. reports on a randomized clinical tria which failed t recruit the goal of 350 patients with advanced HIV infection in order to compare an antiretroviral therapy consititing of either ABC/3TC+DRV/r or ABC/3TC+RAL. In total, onl 46 patients were included, n=22 in the RAL arm and n=24 in the DRV/r arm Thus, the study fails to provide sound and scientifically valid data to make any meaningful conclusions in terms of virological response or other endpoints. Little to nowthing can be said about these combinations for patients presenting late. Same The main conclussion of the authors is that late presenter are frequent but difficult to enroll in clincial trials.

• It is unfortunate that the study failed. At the same time, there is no report in the manuscript about how the study team at least tried to increase patient recruitment during study (proticol changes, increasing the number of study sites etc.)

• The time of recruitment is unclear. In the abstract, the authors state 3 years, in the discussion they state that they did nt expect to have such a low number of patients in 5 years.

Reviewer #4: A two arm non-inferiority randomized study was conducted to compare 48-week virological response of abacavir/lamivudine + darunavir/r vs abacavir/lamivudine + raltegravir in antiretroviral naive late presenter HIV patients. The target sample size was 350; although, due to slow accrual only 46 patients were enrolled. At 48 weeks a virologic success of 77.3% in raltegravir and 66.7% in darunavir/r was observed. Too few patients were enrolled to effectively constructed confidence bounds to test non-inferiority.

Minor revisions:

1- Specify the "parametric or non-parametric tests" that were used.

2- Cite the statistical software used for the analysis.

3- Table 1: In addition to frequencies, provide corresponding percentages for gender, risk factors and CDC stage. For age, HIV RNA, CD4 and CD8 provide the first and third quartiles.

4- Provide confidence intervals for the percentage estimates in the following sentence. “According to a snapshot algorithm, on the ITT population, after 48 weeks the proportion of treatment success was 77.3% in the RAL group and 66.7% in the DRV/r group.”

5- Instead of calculating the IQR, provide the first and third quartiles since they are more informative.

6- Provide numerical p-values rather than “NS.”

7- Include a table summarizing the adverse events by treatment arm. Include percentages corresponding to the frequencies.

6. PLOS authors have the option to publish the peer review history of their article (what does this mean?). If published, this will include your full peer review and any attached files.

Reviewer #1: No

Reviewer #2: Yes: Elbushra Herieka

Reviewer #3: No

Reviewer #4: No

---

## [Author Response · Author response to Decision Letter 0]

13 Aug 2019

Reviewer #1:

1. Discussion, page 18/30, line 3, following, you wrote: “Indeed, due to the low number of subjects enrolled it was not possible to perform any comparative analysis between RAL and DRV/r.” – But you have performed comparisons! And you had interesting findings, e.g. on clinical adverse events and cholesterol differences between groups. – Please clarify.

Answer: we agree with the reviewer and added a sentence in the discussion.

2. Discussion, page 19/30, first lines/first para – please consider to discuss the following facts: in NEAT001-randomized clinical trial (NCT01066962), the raltegravir-containing arm did not perform as well as the standard-of-care arm containing darunavir/ritonavir plus two nucleosidal reverse transcriptase inhibitors, in the subgroup of patients with a CD4 cell count <200/µL. Similar findings were detected in patients with another actual dual regimen in naives: NCT02831764 & NCT02831673.

Answer: we agree with the reviewer and added a paragraph in the discussion.

3. Discussion, page 19/30, lines 14 & following – see above remarks on START study findings release while PRADAR study recruitment.

Answer: we agree with the reviewer and added a sentence in the discussion.

4. Discussion, page 20/30, second para – you showed differences in week 48-outcomes of complete HIV-RNA suppression (<50 copies/mL) between arms, i.e. the primary end point (94.3 vs. 84.2%). Please consider to argue on the differences concerning the 50 copies/mL-threshold for PI vs. INSTI drug classes and to display as well the results for a 200 copies/mL-threshold, as the ACTG recommended this different limit of detection for studies on protease inhibitors.

Actually, no difference compared to <50 copies/mL was found considering a threshold of 200 copies/mL.

5. Figure 1, page 27/30: please consider to include the reasons for 7 screening failures.

We added the reason as footnotes to Figure 1.

Reviewer #2:

1. in Your abstract ( in the last line) you stated: the rate of virologic success is similar to that described in the literature and very far from results of the recent trials in naïve patients. this statement needs to be revisited in my view and quantified.

Answer: we agree with the reviewer and changed the last sentence since it was very vague.

2. in the last line of your introduction: >500000 should be <500000.

Answer: we have corrected the mistake, thanks.

Reviewer #3:

1. It is unfortunate that the study failed. At the same time, there is no report in the manuscript about how the study team at least tried to increase patient recruitment during study (protocol changes, increasing the number of study sites etc.)

Answer: We thank the reviewer and added a sentence in the discussion.

2. The time of recruitment is unclear. In the abstract, the authors state 3 years, in the discussion they state that they did nt expect to have such a low number of patients in 5 years.

Answer: we have corrected the mistake, thanks: the correct recruitment duration was 3 years.

Reviewer #4

1- Specify the "parametric or non-parametric tests" that were used.

The tests are already specified just one sentence after that reported by the reviewer

2- Cite the statistical software used for the analysis.

Done

3- Table 1: In addition to frequencies, provide corresponding percentages for gender, risk factors and CDC stage. For age, HIV RNA, CD4 and CD8 provide the first and third quartiles.

done

4- Provide confidence intervals for the percentage estimates in the following sentence. “According to a snapshot algorithm, on the ITT population, after 48 weeks the proportion of treatment success 

was 77.3% in the RAL group and 66.7% in the DRV/r group.”

done

5- Instead of calculating the IQR, provide the first and third quartiles since they are more informative.

done

6- Provide numerical p-values rather than “NS.”

done

7- Include a table summarizing the adverse events by treatment arm. Include percentages corresponding to the frequencies.

done

---

## [Editor Report · Decision Letter 1]

5 Sep 2019

A Prospective Randomized trial on Abacavir/lamivudine plus DArunavir/ritonavir or Raltegravir in HIV-positive drug-naïve patients with CD4<200 cells/uL (the PRADAR Study)

PONE-D-19-14981R1

Dear Dr. Mussini,

We are pleased to inform you that your manuscript has been judged scientifically suitable for publication and will be formally accepted for publication once it complies with all outstanding technical requirements.

With kind regards,

Alan Winston

Academic Editor

PLOS ONE
---

## [Editor Report · Acceptance letter]

19 Sep 2019

PONE-D-19-14981R1 

A Prospective Randomized trial on Abacavir/lamivudine plus DArunavir/ritonavir or Raltegravir in HIV-positive drug-naïve patients with CD4<200 cells/uL (the PRADAR Study) 

Dear Dr. Mussini:

I am pleased to inform you that your manuscript has been deemed suitable for publication in PLOS ONE. Congratulations! Your manuscript is now with our production department. 

With kind regards,

on behalf of

Prof. Alan Winston 

Academic Editor

PLOS ONE